# Exploring parent-child relationships in families with advanced age parents: An interview study with adult offspring of 'older' parents in Switzerland

Nathalie Bettina Neeser[ID][1]*, Andrea Martani[1], Kato Verghote[2], Bernice Simone Elger[1,3], Tenzin Wangmo[1]

1 Institute for Biomedical Ethics (IBMB), University of Basel, Basel, Switzerland, 2 Department of Philosophy and Moral Science, Bioethics Institute Ghent (BIG), Ghent University, Gent, Belgium, 3 Center for Legal Medicine, University of Geneva, Geneva, Switzerland

* nathalie.neeser@unibas.ch

## Abstract

In many societies, people are becoming parents at an older age. Although there is research on medical implications of advanced parental age (APA), little is known about the parent-child relationships and the lived experiences of offspring with APA parents. This study investigates these topics relying on semi-structured interviews with 20 adult offspring from Switzerland who have at least one parent that was 40 years old or older at the time of their birth. We performed an inductive thematic analysis. After completing the analysis, we found that the themes aligned closely with the theoretical model of intergenerational solidarity and therefore organized our findings in a manner informed by this framework. This allowed us to explore how the four conditional structures of solidarity this model identifies (opportunity, need, family and cultural-contextual) were perceived by our participants and featured in their conceptions of their parent-child relationships. We discuss our findings focusing in particular on the peculiarities that are related to the special dyadic relationship of our participants, i.e., one where parents are particularly old. Given that demographic trends of later-in-life childbearing are likely to continue, the results have implications on family building, (reproductive) ageing, parenthood norms and care arrangements for ageing parents.

## Introduction

In many societies, there is a growing tendency for people to become parents later in life, leading to an increase of advanced parental age (APA). The mean age of first-time mothers in the European Union was 28.8 in 2013, but it has increased to 29.7 in 2021 [1]. In addition, the overall mean age of mothers is continuously increasing and so is the proportion of live births to women aged 40 years or older, which is more than one in 20 as of 2018 [2]. In Switzerland, the mean age of mothers has

**Data availability statement:** All participants have consented to make their data available in an anonymized way on an open repository after the end of the project. The project ends in January 2026, and the data will be curated and deposited in SwissUBase (https://www.swissubase.ch/en/), an open repository in Switzerland that stores and makes available social science data. All readers will have access to the data starting July 2026. The data will be findable on SwissUBase using either the project ID [grant number: 197415] or the name of the authors.

**Funding:** This study is funded by the SNSF-FWO Lead Agency Grant (10001AL_197415 / 1, project title 'Family Building at Advanced Parental Age: An Interdisciplinary Approach') and FWO (Research Foundation – Flanders) (FWO.ORP2021.0001.01). The funder played no role in the design, execution, analysis, interpretation of data, nor in the writing of the study.

**Competing interests:** The authors declared no potential conflicts of interest with respect to the research, authorship, and/or publication of this article.

also steadily increased over the last decades from 27.7 in 1971 to 32.3 in 2021 [3]. This also means that, in 1970, only 11.3 per cent of women were ≥35 years old when giving birth compared to 34.4 per cent in 2021 [4]. A similar trend has been observed in fathers, with the average age of fathers continuously rising over recent years, from 34.0 in 2007 to 35.2 in 2021 [3].

As previous research has demonstrated, the reasons for this trend of later-in-life parenthood among contemporary parents in industrialised countries are manifold, including: the introduction of contraceptive methods; women's level of education and their chosen field of study; women's labour force participation; financial stability; and the perception of the partner's suitability to parent [5,6]. While much research has been conducted on medical complications in connection to advanced maternal age and childbearing [7–9], comparatively little is known about the long-term implications of advanced maternal and/or paternal age on the offspring when they reach adulthood [10]. Previous research indicates that offspring of APA parents may be at a higher risk for adverse cardiovascular health outcomes [11,12], cancers [13,14], schizophrenia [15,16], and autism spectrum disorder [17,18]. But besides this evidence focusing on psycho-medical issues, our knowledge of the psycho-social outcomes of parenthood at advanced age is limited [10,19]. Moreover, there is limited research on the lived experiences of adult offspring of APA parents. The only studies available date back to 1988 and 1991, focusing on the experiences of (adult) offspring to APA parents [20,21].

Whilst (current) studies on the parent-child relationships in APA families are scarce, there is a vast body of empirical research on the parent-child relationship when it comes to caring for ageing parents [22–26] as well as on the influence of siblings on the (shared) care of older parents [27,28]. Consequently, several authors have examined various aspects of the parent-child relationship, including the extent and nature of filial obligations [29]. Previous research has also shown that intergenerational relationships persist into offspring's adulthood (e.g., [30–32]). Szydlik [33] states that such relationships are affected by the societal circumstances parents and their adult offspring live in. He noted that there is an interconnectedness between family and society that goes into both directions: "On one hand, societal contexts influence lifelong intergenerational behaviour. On the other hand, family solidarity may have strong consequences for societies at large." [33, p. 100]. Szydlik condensed this into a theoretical model of intergenerational solidarity [33–35], which constitutes a comprehensive background theory to explore intergenerational relationships (esp. between children and parents). This theory can be used to inform different types of empirical analysis in connection to a variety of intergenerational topics, such as "emotional bonds, help, care and financial transfers" [33, p.102].

This paper is based on an inductive thematic analysis of interviews with adult offspring born from particularly old parents and investigates their views on parent-child relationships. Although the analysis was conducted inductively, the resulting themes showed strong resonance with Szydlik's theory of intergenerational solidarity, which subsequently informed the presentation of the findings in a manner informed by this framework. We therefore focused on exploring whether and/or how the

four conditional structures (opportunity, need, family and cultural-contextual) within Szydlik's theory of intergenerational solidarity apply to parent-child relationships characterized by APA. The focus on this specific type of relationships is very relevant, as parental age continues to increase, and thus a growing number of people may experience these peculiar intergenerational family situations, where children and parents have a substantial (40 + years) age gap.

## Theoretical framework

Szydlik's theoretical model of intergenerational solidarity offers a framework for the analysis of intergenerational relationships in (i) single moments in time; (ii) different snapshots across the course of one's life; and (iii) the investigation of dynamics and changes in intergenerational relations [33]. The model provides a general conceptual framework, and offers a theoretical background and reasoning to be applied as a basis for empirical research and analysis. Szydlik's theory is based on the idea that different facets of solidarity characterise intergenerational relationships. In particular, solidarity has three dimensions: (i) affectual solidarity, meaning the level of emotional closeness between different generations; (ii) associational solidarity, meaning common activities that parents and children do together; and (iii) functional solidarity, meaning the giving and taking of money, time and space between generations. Most importantly, the model underlines that there are four structural determinants that influence the level of intergenerational solidarity (and its dimensions): (i) opportunity structures; (ii) need structures; (iii) family structures; and (iv) cultural-contextual structures. These structures are essential: more or less opportunities (e.g., financial proximities between parents and adult offspring), more or less needs (e.g., health fragility of the older parent), different family structures (e.g., family size) and different socio-cultural contexts (e.g., varying expectations of children's piety) may impact the relationships between different generations. Given the importance of these structural elements, we decided to use them to inform our inductive thematic analysis of the interviews we conducted with adult offspring of APA parents about their experiences and conceptions of the parent-child relationship. In doing so, we wanted to see if there are any peculiarities related to the special dyadic relationship of our participants, i.e., one where their parents are particularly old.

## Methods

### Study design

This study is part of the A-PAGE project, on Family Building at Advanced Parental Age. The project aims to increase insight into the experiences, the moral reasoning and the decision-making processes of "older" parents, professionals and children born when their parents were of an advanced age. The project explores the moral, legal and social significance of age as a factor in these families and in family building in general. To do so, the overall project consists of a systematic review, legal analysis, and a series of explorative semi-structured interviews, including the ones presented here. We received ethics approval from the ethics commission Nordwest- und Zentralschweiz (EKNZ, BASEC-Nr. 2021-01429) and the Ethics Committee of the Faculty of Arts and Philosophy, Ghent University (Ref. 2021-33).

### Sampling and data collection

We sought to find participants aged between 18 and 35, with at least one parent who was 40 years old or older at the time of their birth. The age-range for participants was chosen to identify people who were already young adults, with some of them having vivid memories of their childhood and youth, as they may still be living with their parents or may have moved out recently, and others that may already be confronted with or are increasingly thinking about caregiving tasks for their ageing parents. To maximize the diversity of the relevant characteristics of the sample, purposeful sampling was used to recruit interview participants [36–38]. As we used an explorative qualitative study design, no representative or randomly selected samples were aimed for, but interview participants were recruited until data saturation was reached. More specifically, data saturation was ensured by iterative analysis and reflexive team discussions, deciding whether new interviews

were still generating new insights, or whether thematic saturation was reached. To report the results presented here and to ensure accountability in reporting the study findings, we oriented our reporting on the COREQ Checklist for qualitative research [39].

For the recruitment, we relied on personal contacts and an online-flyer, which was shared on social media platforms (Whatsapp, Instagram) and in an online forum. Most interview participants (n = 17) were recruited through personal contacts of the first author. Although all participants who agreed to be interviewed were asked to suggest other potential participants, as is customary when applying snowball sampling as a recruitment method [38,40], only one participant was recruited through participant referral. Finally, two participants were recruited using an online forum where the study information was advertised.

## Participants

Out of the 20 participants in this study, 13 identified as women, seven as men. Participants were between 19 and 37 years old at the time of the interview. The age of their parents and additional relevant family information are presented in Table 1. The educational level of the participants varied: One had completed secondary school, ten participants had completed an apprenticeship (a practical training in a job, accompanied by studying, lasting three to four years), three held a bachelor's degree, five held a master's degree and one held a PhD. These educational levels reflect the working part of the Swiss population: in 2022 14.3 per cent of the Swiss population's highest educational degree was secondary school, 42.4 per cent had done an apprenticeship and 42.5 per cent had a university degree [41]. At the time of the interview,

Table 1. Participant characteristics.

| Pseudonym and gender | Age category* at time of interview | Mother's age at time of birth** | Father's age at time of birth** | Position of the participant within the family | Second-generation immigrant |
|---|---|---|---|---|---|
| Anna ♀ | 4 | 35 years | 41 years | Oldest child | Yes |
| Benjamin ♂ | 3 | 43 years | 44 years | Youngest child | No |
| Chiara ♀ | 3 | 43 years | 46 years | Youngest child | Yes |
| Daniela ♀ | 4 | 36 years | 43 years | Only child | No |
| Elisa ♀ | 3 | 42 years | 49 years | Youngest child | No |
| Fabienne ♀ | 3 | 34 years | 54 years | Youngest child | Yes |
| Gabriel ♂ | 5 | 44 years | 47 years | Youngest child | No |
| Ivan ♂ | 4 | 38 years | 54 years | Only child | Yes |
| Jennifer ♀ | 1 | 28 years | 47 years | Oldest child | No |
| Kevin ♂ | 1 | 29 years | 41 years | Youngest child | Yes |
| Lukas ♂ | 3 | 34 years | 40 years | Youngest child | Yes |
| Melanie ♀ | 1 | 39 years | 42 years | Oldest child | Yes |
| Nicolas ♂ | 4 | 32 years | 40 years | Youngest child | No |
| Olivia ♀ | 2 | 37 years | 56 years | Youngest child | Yes |
| Patricia ♀ | 3 | 34 years | 43 years | Youngest child | No |
| Ramona ♀ | 1 | 44 years | 36 years | Youngest child | No |
| Sarah ♀ | 2 | 42 years | 51 years | Youngest child | No |
| Thomas ♂ | 3 | 42 years | 28 years | Only child | Yes |
| Ursina ♀ | 2 | 41 years | 38 years | Youngest child | No |
| Vanessa ♀ | 4 | 29 years | 59 years | Youngest child | No |

*To assure that participants were not identifiable, we used age categories to avoid indicating their exact age. All age categories correspond to 4-year intervals. Category 1 = Ages 18–21; Category 2 = Ages 22–25; Category 3 = Ages 26–29; Category 4 = Ages 30–33; Category 5 = Ages 34–37.

**Some of the ages of participant's parents were manipulated to ensure that participants and their families were not identifiable.

19 participants reported being employed, with ten working part-time and nine working full-time. One participant was not employed at the time of the interview as she had recently finished her education. Again, this roughly corresponds to the general population in Switzerland, as in 2023, 86 per cent of women and 94 per cent of men between 25 and 54 reported to be employed [42]. Additionally, in the case of nine participants, their mothers and/or fathers migrated to Switzerland, making nine of the interview participants second-generation immigrants (see Table 1). In 2022, 40 per cent of the Swiss population has had a migration background [43].

## Interviews

Semi-structured, one-on-one interviews were conducted between June 14th and November 7th 2022 either in person (n = 14) or online (n = 6), depending on the preference of the participant. All interviews were conducted by the first author, who informed participants that she is a PhD student interested in their experiences and relevant information in their relationship with their parent(s). The first author, who has a background in social anthropology, was trained and experienced in conducting interviews with study participants on various topics. Study information and an informed consent form were presented to the participants about one week prior to the interview in their preferred language of either English, German or French. After reviewing the study information document and answering any questions, participants provided written informed consent prior to the in-person interviews. For the online interviews, an oral informed consent was obtained before the start of each interview, and a written informed consent was sent by email afterwards. This approach was taken to avoid pressuring the participant in case of technical difficulties.

A semi-structured interview guide with open-ended questions was designed and covered: (a) the participant's experiences with their parent(s) through childhood, adolescence and adulthood and how their relationships with their parents changed over time; (b) their views and personal and social experiences with having (an) older parent(s); (c) their moral reasoning in relation to their own role and that of others in starting a family at an advanced age; and (d) their potential moral concerns in connection to family building at advanced parental age. Additionally, in order to facilitate the data collection on moral reasoning, and to encourage participants to engage with the subject matter in a more abstract manner, a specifically designed elicitation technique was used [44], presenting participants normative statements regarding ageing and family relationships (e.g., "When a parent needs care themselves, their child(ren) can be expected to take care of that parent" or "Imagine a single parent with health problems. Their child of 35 and/or 15 years can be expected to do something for that parent"). The participants were instructed to freely explore if (and under which circumstances) they would (dis)agree with the specific statement. These statements along with other data from the interview formed part of the data relevant for this paper, as they are particularly relevant for our research question focused on the conception of parent-child relationships as viewed by children of APA parents. Parts of the interview guide were pilot-tested with one participant before recruitment started for this study. No repeat interviews were conducted.

Of the 20 interviews, 19 were conducted in (Swiss) German and one was conducted in English. Interviews ranged between 52 and 158 minutes, with an average of 95 minutes. Field notes were taken during and after the interviews to (a) remember potential follow-up questions and (b) to reflect on the interview situation and to improve the interviewing technique. The interviews were recorded (audio) and transcribed verbatim. All identifiable information, such as names and places were anonymised. Transcripts were not returned to participants for their comments, but directly analysed without their further input.

## Analysis

For this paper, we relied on applied thematic analysis [45]. We started with an inductive analysis, but noted that the resulting themes aligned very closely with the model of intergenerational solidarity by Szydlik, which then provided us with heuristic categories that we used to explore the data further (i.e., the four structural determinants of intergenerational solidarity). Concretely, the analysis was aided by three auditing meetings [46] led by the first author. Auditing meetings

involved a presentation of representative quotes of data selected by the first author and informed by the theoretical framework selected for this study. Then, co-authors started discussing the quotes, their interpretations based on the theoretical framework, and their significance for the research question. These auditing rounds aimed to challenge, adapt and refine the thematic structure of the results, and to validate the initial analysis proposed by the first author. The different academic backgrounds of all co-authors – social anthropology, gender studies, bioethics, gerontology, legal analysis, and medicine – allowed for an interdisciplinary approach, strengthening and deepening the final thematic structure.

Following the inductive coding process, we sought to thematically structure the codes using Szydlik's structural determinants of solidarity, that are part of the model of intergenerational solidarity [33–35]. The auditing rounds allowed us to adapt Szydlik's four determinants of solidarity based on the content of our interviews and quotes. The first determinant "cultural-contextual structures" became "historical developments and cultural-contextual structures", and the second determinant "family" became "the specific family structure(s) in place". Instead of the "needs" and "opportunities" as general terms, we chose to present them in a more descriptive way, to represent the most representative needs and opportunities that we identified in our interviews. These were, respectively, "early confrontation with multiple simultaneous burdens", and "offspring's awareness of less "overlapping" time with one's parents".

## Results

In accordance with the aim of this paper, we inductively analysed our interviews. The analysis closely aligned with Szydlik's structural determinants of solidarity (opportunity, need, family, cultural-contextual structures) from his theory of intergenerational relationship [33–35]. Our findings are presented below and summarised in Fig 1.

### Historical developments and cultural-contextual structures

Szydlik's model of intergenerational solidarity proposes to consider the historical developments and the cultural-contextual structures in place to understand parent-child relationships. When focusing on the geographical context of Switzerland and on the experiences of the study participants, two historical and socio-cultural aspects came to the fore in the interviews. First, Switzerland's own history from a predominantly rural to an increasingly urban country has profoundly defined

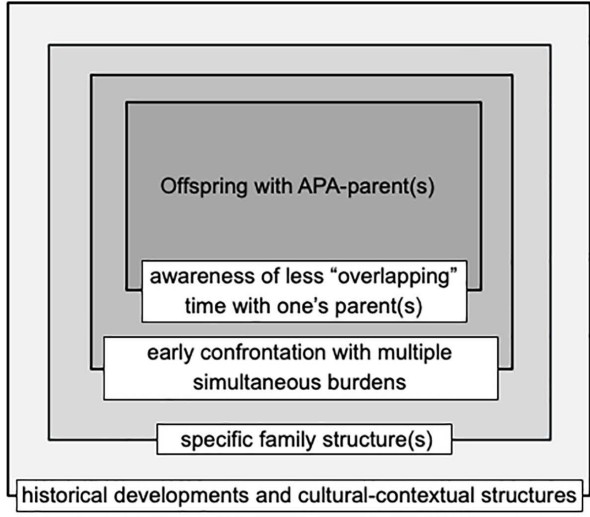

**Fig 1. Thematic structure.**

the way family structures are understood, as exemplified by Nicolas. He reflected on the development of traditional family structures in the rural context, and how this is challenged by modern life, thus shaping the care for older generations and the interaction with them:

> "200 years ago it was somehow all farming families and then it was clear: […] Everyone is together on the Alp or on the fields in the lowlands. And somehow, you bring the different generations along with you. And these family structures allow you to do so, until old age. And today, where society, or respectively our economy almost demands that only one parent or family member goes to work, or meanwhile also two, it is no longer possible to do it [taking care of older family members] this way." (Nicolas)

Nicolas underlined the country's current economic situation, in which it is no longer possible for one person to stay at home and to take care of children and/or older generations. The same aspect was also addressed by Patricia, but from a different angle. She mentioned that being a part of a well-paid workforce and living in a country with a well-developed healthcare system in place affects the interactions with (and the care for) the older generation(s). She states, that children might spend less time with their parent(s), but that they also have the opportunity to afford buying care for older persons in nursing institutions.

> "I mean, we are in a country where we can sort of afford to "give" parents away to a retirement home. Ehm, while in other countries it is like that, that the children are like an old-age pension." (Patricia)

The second historical and socio-cultural element that respondents mentioned interplaying with their parental relationship is migration. Indeed, Switzerland is a country with a rich migration history, which influences perceptions about family traditions. This diversity is also reflected in our sample, as several of the participants were second generation immigrants to Switzerland (see Table 1). The cultural differences and how these shape one's thought processes and one's views on caregiving for an APA parent were discussed by Chiara, who reflects on a very gendered view of caregiving in her parent's country of origin:

> "This is a model that is lived in the global south in particular. Where the [word for 'grandma' in her language of origin] then really moves in at home. And in [name of the country], it is also normal for the youngest female sibling to simply look after her grandmother. So she's not being married either, or she's not getting married most of the time, she's just looking after the grandmother. And that is already a big obstacle, or burden [for the youngest female family member]." (Chiara)

It is in a different way that Ivan talked about his parents being first generation immigrants in Switzerland. In fact, he is uncertain if he will be able to take care of his parents at all, as they are considering to spend their remaining years in their country of origin, whilst Ivan sees his future in Switzerland. He shows awareness that these may be their last years and is worried about how their relationship might develop in the future:

> "Maybe they won't be here [in Switzerland] anymore. Maybe they'll say, no, we'll spend the rest of our lives in [name of the country of origin]. […] And of course, I also support them in this and I also want that for them. That they still have the last few, hopefully many more years, but just that they then have more sun and classical vacations. Yes, but that actually leads to the fact that we will distance ourselves from each other." (Ivan)

Because his parents planned to repatriate, Ivan expressed worry about both their physical and emotional proximity. One's migration background, the historical developments and the wealth of Switzerland has profoundly shaped the

experiences of interview participants' (potential) caregiving roles towards their APA parents. They talked about economic possibilities to buy care for their ageing parent(s) when they themselves cannot provide hands-on care. Also, perceiving the cultural obligation to take care of their parents and not living in the same country makes fulfilling such responsibilities in person impossible.

## Specific family structure(s)

In a next step, Szydlik's model of intergenerational solidarity focuses on family structure(s), a theme that was also strongly reflected in our data. Accordingly, we present the specific relevant family structures in place and how they shape the parent-child relationship in APA families: (a) having at least one parent of APA; (b) the age of a child in question at a specific moment in time when, e.g., caregiving tasks are increasing; and (c) the quality of the past parent-child relationship – for example while growing up – which dictates the relationship in the present and/or the future.

For the first aspect, all study participants shared that they all had at least one parent that was 40 years old or older when the participants were born. As parents had their child(ren) at a later point in time compared to the mean parenting age in Switzerland, participants reported being confronted with feelings of their family situation being different in comparison to their peers. While Anna, for example, described her parents as from "a different generation", Sarah perceived her family situation to be "different" due to her having much older half-siblings, and Ramona expressed that her family felt different from her friends' because her mother was older than the other mothers. Interestingly, Ramona, as other participants, sometimes turned things around and assigned the label of "normal" to their family situation, thus making others' family the "different" ones.

For the second aspect, most participants expressed that there was a considerable difference in their views towards caregiving responsibilities of APA children depending on their age. Children of APA parents are more likely to face a situation where they have to care for their parents earlier in life, since the age difference with their parents is higher and these may need help when their children are still very young. In this respect, participants voiced their concerns that they did not think that someone should be tasked or feel obliged to take care of ageing parents too early in their life, for example at 15 years old. Sarah expressed this very clearly, focusing on the importance of the child being able to 'be a child' and to focus on the experiences that society, would ascribe as fitting for a child of that specific age group:

*"I sometimes find it a bit difficult, generally how much is expected of children somehow. And a 15-year-old is simply still a child. And I think it's much more important that a child is allowed to be a child." (Sarah)*

Sarah was not alone in expressing this opinion. Both Anna and Vanessa strongly voiced that a child is the vulnerable individual in the parent-child relationship and should therefore not take on a lot of responsibility too soon. Anna said that "[With] 15 [one is] still a child, they need care first" and Vanessa took a strong stand when she said "You cannot be expected to somehow take on responsibility […] for a parent". Ursina similarly expressed that protecting the child and their needs should come first, but she went a bit further when reflecting about the child's wishes to help. Depending on the relationship to one's parent(s), the child might have a close emotional bond to the parent and may actively express the wish to help. In this situation, Ursina stated that it is important that the child is not confronted with the situation on its own, but that there is an adult around, measuring how much might be "too much" and never leaving the responsibility on the child's shoulders alone:

*"The child is allowed to help, the child is allowed to experience it, but it must have the opportunity to withdraw and say: Hey, this is all too much for me. Even if it's my mom or even if it's my dad, I have to take a step back. And this child will never understand this by itself. This child will never say on its own: I can't be there for my mom. But the child will always have this burden on its shoulders, that it was not there, because it's mom or it's dad. That means there has to*

                                                                                                                                                           

*be an adult who says: Hey, we're going to take you to the side and someone else is going to do it. Because otherwise it's not good at all, neither for the parent nor for the child." (Ursina)*

While most participants expressed that with 15 a child is too young to take on care responsibilities and should themselves be protected and cared for, Daniela had a slightly different approach. Even though she prioritized the care an offspring receives from their parent(s) and makes clear that the satisfaction of the child's needs is essential, she also expressed the importance of the child helping their parent(s) as a means for the parent(s) to then be able to take care of the child in return:

*"Of course you're only 15 years old. And that is really a burden. […] But I think, how else would it work? So you live, well, if you live at home and then you do not help, how can you be there? I think then you almost cut a bit into your own flesh. Because the parent takes care of you as well and if you don't take care of your mom or dad, they can't take care of you anymore." (Daniela)*

As Daniela pointed out, children are highly dependent on their parents. But as is the case with the connection between society and the intergenerational behaviour in families [33], it is the same when it comes to the dependence of a young child and a parent needing care: it does not only go into one direction, but in both.

For the third aspect, the importance of a good and well-balanced parent-child relationship in the present, several participants stressed the importance of how well one's former relationship with a parent was, for example while growing up or while going through challenging times in one's life. In fact, when asked about potentially caring for a parent, both Ursina and Olivia felt strongly about the influence of one's relationship with a parent in the past and specifically pointed out under which circumstances there might be reciprocity – and under which ones not:

*"There are always situations or family constructs where, for example, you can't handle it when you're around this person. Because there's just a lot that's already happened, or there's just things that have happened in the background. I mean, if a child now for example grows up in a household where the parents are violent or something like that, then I can understand if this child, when they are adult, says: Hey I don't want to take care of my parents." (Ursina)*

*"If it's a family in which you have a really shitty relationship, and it never worked out from an early age onwards and the parents were never there for the child, then you can't expect the child to be there for you when you weren't there yourself." (Olivia)*

## Early confrontation with multiple simultaneous burdens

Informed by Szydlik's conception of "need structures", we identified in our interviews one specific set of needs that participants often used to characterise their specific situation of being children of particularly older parents. This is the early confrontation with multiple burdens at the same time, meaning that participants expressed having to care (earlier) for the needs of their own (older) parents and simultaneously for their own personal (and potentially their family) needs. Participants had a look back at their childhood memories and pointed out how their parents did not have the capacity to walk around with them anymore (Fabienne), or that they were the ones steadying their father on icy roads instead of being steadied by their parent (Vanessa), but also reflected on the present or had a look towards the future concerning their care responsibilities. Vanessa, for example, also pragmatically highlighted the multiple burdens in questions, when saying that someone might be "fully occupied in a job or you have three children at home or something, then you just can't go to your mother or father all the time" (Vanessa). Additionally, Anna reflected on her parent's APA in connection with her personal timing and decision-making in relation to family building:

*"I'm at this stage where anyway, if I have children, it will be late. So [my parents] won't be able to help me in caring for [my children]. And I would be split in half, because I would have to care for the elder and for the younger. That becomes difficult to manage. [...] But even if I don't have children, the fact is that I still have to juggle with my personal life and career and at the same time care for my older parents, when most of my friends around me don't have that element." (Anna)*

Anna clearly felt that she was spreading herself thin as she questioned how her own family planning might differ from her peer's due to her parents' age. Increasingly fulfilling caregiving tasks for her older parents and simultaneously being confronted with other potential time- and energy consuming tasks is not the only aspect to it. Rather, she also points out the discrepancy in comparison to her peer's experiences when it comes to her thoughts on building her own family. While her peers who have younger parents might be less confronted with difficulties in finding competent childcare for their off-spring (as their parents are engaged in caregiving tasks for their grandchildren), she partly felt deprived of the experience of her parents being (involved) grandparents in their grandchildren's lives:

*"Due to the fact, that since they're already quite older, they won't be very present grandparents. [...] If we ever have children, they won't have grandparents for a very long time. And maybe even if, let's say they stayed on, they would be grandparents who wouldn't be able to do so many things with the children [due to their old age and health ailments]. So, because many people actually count on their parents to take care of their own children. The older you get your kids, and if your parents had you older, then that, you can't really count on that model anymore to have the grandparents help in caring for the children. [...] I think that's a bit of a shame. But it's also partly their fault. They had me late and if I have children late, well that's it." (Anna)*

Study participants also often pointed out that they have not had the experience of having grandparents themselves. The decision of the APA parents to have (a) child(ren) at 40 years old or older shaped not only their relationship with their own children, but also two additional ones. The children of APA-parents' (i.e., the one in our sample) rarely interacted with their own grandparents, and they often expressed that their own children would not have the chance to meet their grandparents.

## Awareness of less "overlapping" time with one's parents

The last conditional factor shaping the parent-child relationship according to Szdylik are opportunity structures. These are all those factors that "reflect opportunities or resources for solidarity" and "enable, promote, hinder or prevent social inter-action" [33]. In our data, one prominent opportunity structure mentioned by children of APA parents was their awareness that the time they could share with their parents (in the sense of 'both being alive and healthy') is limited, thus impacting social interaction in several ways. As we have previously described, some participants expressed uncertainty about their APA parents being potential grandparents and their abilities to support them with childcare tasks for grandchildren. However, other participant's doubts and uncertainties even went a step further. Most participants also expressed awareness that they would not have their parent(s) as long in their lives as their peers who have younger parents. They were worried that they might not have their parents in their lives as long as they wished for and anticipated the early loss of a parent. One participant described wanting to marry early as a way to ensure her father being able to walk her down the aisle and in the case of another participant, this led to reflections about how the parent-child relationship will evolve in the future. Gabriel described:

*"I assume that their need for support will increase. That we will have to look out for them more and help out a little bit. What is probably also going to happen is, that somehow spending time together will become more valuable and that we*

*will live more consciously. Even just when you visit them, that is something special for them: that you see each other again, that you talk a bit together and that you go into the garden together and so on." (Gabriel)*

As Gabriel put quite clearly, he expected the appreciation of the time spent together to increase in the next years and to live in a more conscious way with his parents. Several other participants similarly demonstrated conscious reflection about the time they spend with their parents in the present and the future. As reflected in the following quote, Nicolas described the time with his father as finite:

*"And I think that has now started in the last five years or so, that I started having this awareness that he probably will not live to be 100 years old. Already with his health problems. So you somehow have to use this time as much as possible." (Nicolas)*

Vanessa described a similar situation in the past. She realized very early on that her father will at some point "no longer be here". Consequently, she started living more consciously and with a heightened awareness that she only has a limited time with him. She retrospectively talked about the finiteness that shaped the relationship with her father, as her father had already deceased two years prior to the interview. She explained:

*"I actually realized 15 years earlier: At some point, he will no longer be here. And at that time, I surely cried a lot and I also got a bit carried away sometimes. But I have to say, that I'm very glad about this. Because in these 15 years, I'm very, very glad, that I did what I did. Because I actually lived extremely consciously with him through that, or spent time with him. And I was actually very aware: I won't have that until I'm 40." (Vanessa)*

While Vanessa's peers might have had the time to focus on their personal goals and ideas of how their life is going to look like – especially in these very formative years during one's teens – Vanessa knew very well how her future would not look like and that she would have to come to terms with losing her father at an early time in her life. Several other participants reflected on the influence of the awareness that one's time with a parent might be shorter in comparison to what their social environment and their peers with younger parents might experience.

## Discussion

In this study, we presented findings of how adult offspring with at least one APA parent view their parent-child relationships. We explored, in particular, whether and how the theory of intergenerational solidarity, and its conceptions of conditional factors shaping parent-child relationships [33–35] applied to experiences of adult offspring with older parents. While the intergenerational solidarity model was previously used to explore topics such as "emotional bonds, help, care and financial transfers" [33, p. 102], we applied it as a theoretical background to explore how a specific (but increasingly important in absolute numbers) group of adult offspring perceives the relationship with their parents.

When it comes to the influence of historical developments and the cultural-contextual structures, our study participants confirmed the influence of these elements on how they perceived their relationship with their APA parents. In the specific context of Switzerland, the gradual move away from traditional family structures in the rural context posed challenges to how to redesign care responsibilities between generations. Moreover, in comparison to other countries, Switzerland is relatively wealthy [47], meaning care is increasingly perceived as a commodity. That is, when children do not have time to provide care to ageing parents, it is relatively easy to outsource it due to a well-developed healthcare system [48]. Third, Switzerland has a long-lasting migration history and there is not a single and universal conception of what makes a family and how the relationship between parents and children should look like. Currently, 40% of the resident population has a migration background [49], a percentage that reaches 60% if children alone are considered [50]. Therefore, the realities of

the parent-child relationships described are constantly formed and re-formed over time through many socio-cultural influences. These results align with existing gerontology literature, where studies that discuss caregiving relationships among immigrant families are abundant (e.g., [51–53]). However, they are important to mention, as the three aspects form the base line for the discussed determinants in the following, and also make apparent that families with (an) APA-parent(s) do not differ from other families in that respect, but face similar challenges.

The specific family structure(s) play(s) an essential role in the parent-child relationship according to Syzdlik theory, and we found evidence of this in our data. Naturally, one common trait of all family structures of the participants in this study was that all had at least one parent being 40 years old or older at the time of their birth. For many of the participants, this involved that the offspring were confronted with their parents' ageing, deteriorating health and care-related activities earlier in their life compared to their peers. Our results allowed to explore their moral reaction to this reality: does being confronted with a situation of need create necessarily a feeling of moral duty to address it? Our results show that two factors are perceived as '(de)activating' the moral duty to care. First, the age of the offspring when they face potential caregiving tasks for an ageing parent was deemed decisive. Participants were generally outspoken against the presence of a strong moral duty to care for older parents in cases where this burden would fall on a child or someone in their early teenage years. On the contrary, participants supported that it is morally acceptable for an adult child to shoulder greater care responsibilities for an ageing parent. Although previous research has shown that intergenerational relationships persist into adulthood [30–32], the intergenerational relationships in APA families might differ profoundly from family structures where parents are younger. In APA families, younger children are more likely to be in their teenage years when parents experience the signs of ageing and their needs increase. Second, participants stated that the past parent-child relationship, for example while growing up, is significant to decide whether the child has a moral duty to care for the ageing parent in the future. Participants were outspoken about the fact that if a parent had provided a caring household while growing up, they thought that there is a stronger moral duty to look after the parent. However, if the relationship in the past had been poor and the offspring recounted not being properly cared for, they disagreed that children have a strong moral duty to care for their parents in times of need. This corresponds to one of the theories of filial obligation identified by Stuifbergen and Delden [29], reciprocity, whereby moral obligations of children towards their parents should mirror the level of care they received from them when young. However, reciprocity is not the only motivation for filial obligations, as there is also research speaking about children taking care of ageing parents despite suboptimal care received in the past [54,55]. Moreover, it is important to underline that most of the participants in the current study were not at the stage of care provision for their parents. It is possible that their actions will differ from their current opinions at a time when their parents are in need of care.

Our results indicated that our participants experienced an early confrontation with multiple simultaneous burdens due to the age of their parents. To be multiply burdened per se is nothing new and has been extensively studied under the terminology "sandwich generation" (e.g., [56–58]). Additionally, there has been considerable research when it comes to young (adult) carers in the context of Europe and Switzerland [59–61]. Although young carers (up to 18 years) and young adult carers (18–24 years) often do not identify themselves as such, they nonetheless provide care to (a) family member(s) and their needs are often overlooked [59,60]. The experiences of offspring with (an) APA-parent(s) seem to be potentially located at the intersection of the "sandwich generation" and the "young (adult) carers" as results show that offspring to APA-parent(s) might be subjected to this aspect of being multiply burdened at a considerably earlier time in their lives. In fact, some participants relayed their experience of having a role reversal quite early. That is, they reveal, having to take care of their parents instead of being the ones taken care of quite early in their lives. In the case of some participants who had not yet faced role reversals, they worried about how their relationship to their parent(s) would change in the future, when their parents are in increasing need of care at a point in life when they may not feel ready to adequately provide it. Future research should explore whether our exploratory results also stand in larger samples and should control for other variables, e.g., family size, especially in cases of parents with an only child.

Finally, when it comes to the offspring's awareness of less "overlapping" time with their parent(s), we found that some participants were aware of their parents' advanced age quite early in their lives and consequently also became aware that they would potentially be confronted with an early loss of their parent and that their children would potentially never meet their grandparents. Although the existing literature on offspring to APA parents suggested that this might lead to fear or even remorse when it comes to the parent's decision-making to have child(ren) at a later point in live [20,21], we did not identify strong feelings of fear or remorse in the participants. In fact, the participants exhibited a heightened awareness of the potential for their parents to die in the imminent future. This realisation contributed to the participants' appreciation of the time that they were still able to spend with their parent(s). Previous studies on APA have shed light on the potential for offspring to develop two types of fears: that of their parents' premature death and that of their own child not having a grandparent [62]. Our results only support the presence of latter fear. Regarding the former, our results demonstrate the opposite: While some participants expressed sadness about the perceived limited time they had left with their parents, they did not attach feelings of fear to this realisation. Rather, they described feeling grateful for the time they had left with their parents and expressed a sense of readiness to eventually bid them farewell.

### Strengths and limitations

Important limitations are that as a qualitative study, the findings are not generalizable. Offspring of APA parents elsewhere may have completely different viewpoints and experiences. Also, it is possible that offspring with even older parents (e.g., 50+ at the time of becoming parents) might describe different parent-child relationships than our participants. However, we decided to examine the experiences of children born to 40+ parents, since this age was – at the time when our participants were born – much more exceptional than it is nowadays. Moreover, existing literature on the phenomenon is very limited, and previously defined APA as 35 years and older [20,21], so our decision to recruit children born from parents aged 40+ is already a step forward in adapting the definition of APA to demographic changes. Additionally, our sample is self-selected and thus, it is possible that offspring with exceptionally negative experiences and/or views of APA might not have considered to participate. Also, in our study we have solely focused on the experiences of offspring, but have not interviewed parent and child dyads, which could in the future potentially lead to fruitful results. Finally, to study advanced parental age and to recruit potential interview participants, we have chosen to define APA as a parent that is 40 years old or older. In the literature, there is not one single definition of APA, but a variety of definitions is present [10]. With the demographic development of parental age rising, it is likely that the definition of APA as 40 years old or older might not be applicable anymore in the future. However, our findings offer a baseline of an understanding of offspring's experiences and views of APA and of the parent-child relationship in APA families.

One of the biggest strengths of this study is its examination of a steadily growing, but understudied group, that is, offspring of advanced age parents and the parent-child relationship in such families. The use of an explorative study design and the specifically designed elicitations techniques have provided valuable insights into the offspring's moral reasoning and their more abstract views on filial duties. When it comes to the sample, the most important demographic characteristics of people living in Switzerland have been portrayed, as we interviewed people of different ages and with different socioeconomic backgrounds.

### Conclusion

Our study offers insights into the parent-child relationship in families with parents of APA, focusing in particular on the four conditional determinants of intergenerational solidarity of Szidlik's theory. Since APA is a growing phenomenon, it is increasingly important to understand not only how it impacts the somatic health of children and parents around birth, but also what the repercussions are (if there are any) on the long-term family functioning and the intergenerational parent-child relationship. By analysing the lived experiences and views of adult offspring born to particularly old parents, we have provided some initial insights on how their specific situation interacts with structural elements impacting

intergenerational relationships. Future research with larger samples and quantitative designs should continue to examine these topics, in particular to help develop reproductive and family policies that are informed by a more complete picture of what having children at a later stage in life entails.

## Supporting information

**S1 Table. COREQ (COnsolidated criteria for REporting Qualitative research) Checklist.**
(DOCX)

## Acknowledgments

Our sincere gratitude goes to all the interviewees for their time and willingness to participate in an interview and share their experiences. We also thank the Swiss-Belgian A-PAGE collaborative team (Prof. Dr. Veerle Provoost, Prof. Dr. Guido Pennings and Dr. Steven Piek) for their valuable comments on this paper and their general contributions towards the project.

Portions of an earlier version of this manuscript were presented at the 2023 European Association of Centres of Medical Ethics (EACME) Conference in Warsaw, Poland.

## Author contributions

**Conceptualization:** Nathalie Bettina Neeser.

**Data curation:** Nathalie Bettina Neeser.

**Formal analysis:** Nathalie Bettina Neeser, Andrea Martani, Kato Verghote, Bernice Simone Elger, Tenzin Wangmo.

**Funding acquisition:** Bernice Simone Elger, Tenzin Wangmo.

**Supervision:** Tenzin Wangmo.

**Visualization:** Nathalie Bettina Neeser.

**Writing – original draft:** Nathalie Bettina Neeser, Andrea Martani, Kato Verghote, Bernice Simone Elger, Tenzin Wangmo.

**Writing – review & editing:** Nathalie Bettina Neeser, Andrea Martani, Kato Verghote, Bernice Simone Elger, Tenzin Wangmo.

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
