## [Decision Letter · Decision Letter 0]

17 Nov 2025

Dear Dr. Neeser,

Thank you for submitting your manuscript to PLOS ONE. After careful consideration, we feel that it has merit but does not fully meet PLOS ONE’s publication criteria as it currently stands. Therefore, we invite you to submit a revised version of the manuscript that addresses the points raised during the review process.

We look forward to receiving your revised manuscript.

Kind regards,

Adetayo Olorunlana, Ph.D.

Academic Editor

PLOS ONE

2. In the online submission form, you indicated that [The data that support the findings of this study are available from the corresponding author upon reasonable request.].

Additional Editor Comments (if provided):

Reviewers' comments:

Reviewer's Responses to Questions

**Comments to the Author**

1. Is the manuscript technically sound, and do the data support the conclusions?

Reviewer #1: Yes

Reviewer #2: Yes

2. Has the statistical analysis been performed appropriately and rigorously?

Reviewer #1: I Don't Know

Reviewer #2: Yes

3. Have the authors made all data underlying the findings in their manuscript fully available?

Reviewer #1: Yes

Reviewer #2: Yes

4. Is the manuscript presented in an intelligible fashion and written in standard English?

Reviewer #1: Yes

Reviewer #2: Yes

Reviewer #1: This article is very interesting and I like it very much. It's quite different from the ones I have reviewed before. However, I think it leans more towards the field of psychology. I suggest having it reviewed by an expert in psychology. Thank you!

Reviewer #2: This manuscript addresses an important and understudied phenomenon—the lived experiences of adult offspring born to parents of advanced parental age (APA). Comments are below:

-The authors claim "inductive thematic analysis" but acknowledge starting with Szydlik's predetermined categories (opportunity, need, family, cultural-contextual structures). This is actually deductive or framework-based analysis. Cab you please clarify whether you used a deductive approach with some inductive elaboration, or explain how themes genuinely emerged from data before being mapped to theory?

-Given rising parental ages globally, is 40 still meaningfully "advanced"? Can you please clarify? I'm curious if you could add discussion in the limitation section on your findings whether findings differ for parents at 40-43 vs. 50+ years?

- How do the team ensure data were saturated?

-Some participants (e.g., Anna, Nicolas, Vanessa) are quoted extensively, while others appear only once or not at all. This raises questions about whether all 20 participants contributed equally to theme development or if findings are driven by a subset.

**Do you want your identity to be public for this peer review?** For information about this choice, including consent withdrawal, please see our Privacy Policy

Reviewer #1: No

Reviewer #2: No

---

## [Author Response · Author response to Decision Letter 1]

12 Dec 2025

05.12.2025

Dear Editor,

Thank you for your consideration of our revised manuscript. We are pleased to submit a new version of our manuscript titled “Exploring parent-child relationships in families with advanced age parents: an interview study with adult offspring of ‘older’ parents in Switzerland” to PLOS ONE. Based on the much-appreciated feedback and the helpful suggestions of the two reviewers, we have carefully considered the valuable comments and adapted the manuscript where necessary accordingly. Below, we copy-pasted all the comments of the reviewers and explained how their comments have been implemented in the revised version with track changes.

We thank you and the reviewers for raising important issues and helping us refine the manuscript.

With sincere wishes,

The Authors

*****

Responses to the Reviewers

Reviewer #1

This article is very interesting, and I like it very much. It's quite different from the ones I have reviewed before. However, I think it leans more towards the field of psychology. I suggest having it reviewed by an expert in psychology. Thank you!

Authors’ response: We would like to thank Reviewer #1 for taking the time to read our manuscript and are pleased to hear that they find it interesting.

Reviewer #2

This manuscript addresses an important and understudied phenomenon—the lived experiences of adult offspring born to parents of advanced parental age (APA). Comments are below:

1. The authors claim "inductive thematic analysis" but acknowledge starting with Szydlik's predetermined categories (opportunity, need, family, cultural-contextual structures). This is actually deductive or framework-based analysis. Can you please clarify whether you used a deductive approach with some inductive elaboration, or explain how themes genuinely emerged from data before being mapped to theory?

Authors’ response: Thank you for bringing up this important matter. We have specified what we have done exactly; in the abstract, the introduction and in the methods section of our manuscript. For example, in the introduction, we had previously written that “this paper is based on applying Szydlik’s theory of intergenerational solidarity to analyse interviews with adult offspring born from particularly old parents and investigates their views on parent-child relationships” and have now specified this part of the last paragraph of the introduction as follows: “This paper is based on an inductive thematic analysis of interviews with adult offspring born from particularly old parents and investigates their views on parent-child relationships. Although the analysis was conducted inductively, the resulting themes showed strong resonance with Szydlik’s theory of intergenerational solidarity, which subsequently informed the presentation of the findings in a manner informed by this framework.”

2. Given rising parental ages globally, is 40 still meaningfully "advanced"? Can you please clarify? I'm curious if you could add discussion in the limitation section on your findings whether findings differ for parents at 40-43 vs. 50+ years?

Authors’ response: Thank you for this important comment. We have added several sentences to our strengths and limitations section, detailing that “it is possible that offspring with even older parents (e.g. 50+ at the time of becoming parents) might describe different parent-child relationships than our participants. However, we decided to examine the experiences of children born to 40+ parents, since this age was – at the time when our participants were born – much more exceptional than it is nowadays. Moreover, existing literature on the phenomenon is very limited, and previously defined APA as 35 years and older [20,21], so our decision to recruit children born from parents aged 40+ is already a step forward in adapting the definition of APA to demographic changes.”

3. How do the team ensure data were saturated?

Authors’ response: Thank you for raising this important issue in qualitative research. We have specified our process in the methods section of the manuscript, detailing that “More specifically, data saturation was ensured by iterative analysis and reflexive team discussions, deciding whether new interviews were still generating new insights, or whether thematic saturation was reached.”

4. Some participants (e.g., Anna, Nicolas, Vanessa) are quoted extensively, while others appear only once or not at all. This raises questions about whether all 20 participants contributed equally to theme development or if findings are driven by a subset.

Authors’ response: You are correct, that some participant’s wordings and detailed explanations are displayed and discussed more often than the one of others. However, the findings are not driven by a subset, but rather by the sample in its entirety. We chose to display these participants’ words in particular, because they reflected in depth and at length about different issues and articulated themselves exceptionally well. However, all participant contributed to the development of the findings.

*****

Journal requirements

In the online submission form, you indicated that [The data that support the findings of this study are available from the corresponding author upon reasonable request.].

Authors’ response: In the consent forms that participants signed, we asked them to sign twice, once for the use of the interview data within the project and once to put their anonymized data on an open repository after the end of the project. Since not everyone decided to consent to having their data put on an open repository and the project has not yet come to an end, the data that support the findings of this study are available from the corresponding author upon reasonable request. We take informed consent very seriously.

---

## [Decision Letter · Decision Letter 1]

21 Dec 2025

Exploring parent-child relationships in families with advanced age parents: an interview study with adult offspring of ‘older’ parents in Switzerland

PONE-D-25-33579R1

Dear Dr. Neeser,

We’re pleased to inform you that your manuscript has been judged scientifically suitable for publication and will be formally accepted for publication once it meets all outstanding technical requirements.

Kind regards,

Adetayo Olorunlana, Ph.D.

Academic Editor

PLOS One

Additional Editor Comments (optional):

Reviewers' comments:

Reviewer's Responses to Questions

**Comments to the Author**

Reviewer #2: All comments have been addressed

2. Is the manuscript technically sound, and do the data support the conclusions?

Reviewer #2: Yes

3. Has the statistical analysis been performed appropriately and rigorously?

Reviewer #2: Yes

4. Have the authors made all data underlying the findings in their manuscript fully available?

Reviewer #2: Yes

5. Is the manuscript presented in an intelligible fashion and written in standard English?

Reviewer #2: Yes

Reviewer #2: (No Response)

**Do you want your identity to be public for this peer review?** For information about this choice, including consent withdrawal, please see our Privacy Policy

Reviewer #2: No

---

## [Editor Report · Acceptance letter]

PONE-D-25-33579R1

PLOS One

Dear Dr. Neeser,

I'm pleased to inform you that your manuscript has been deemed suitable for publication in PLOS One. Congratulations! Your manuscript is now being handed over to our production team.

Kind regards,

on behalf of

Associate Professor Adetayo Olorunlana

Academic Editor

PLOS One